# Coffee Intake in Brazil Influences the Consumption of Sugar, Sweets, and Beverages

**DOI:** 10.3390/nu16234019

**Published:** 2024-11-24

**Authors:** Marijoe Braga Alves Simões, Joana Maia Brandão, Anna Beatriz Souza Antunes, Rosely Sichieri

**Affiliations:** Social Medicine Institute, University of the State of Rio de Janeiro (UERJ), Maracanã 20550-900, Rio de Janeiro, Brazil; joanamaia24@gmail.com (J.M.B.); antunes.abnutri@gmail.com (A.B.S.A.); rosely.sichieri@gmail.com (R.S.)

**Keywords:** coffee consumption, sugar, non-caloric sweeteners, sugary food

## Abstract

**Background/Objectives:** Considering the high consumption of coffee in Brazil, this study aimed to investigate the relationship between coffee consumption and the intake of added sugar, non-caloric sweeteners, sugary beverages, and foods. **Methods:** A modified case-crossover study was conducted using data from the national “Household Budget Survey (POF)” which 38,854 participants. Dietary intake was assessed using 24-h recalls on two non-consecutive days. Days with and without coffee consumption were compared (case-crossover) to evaluating the effects on sugar, sweeteners, sugary foods, sugar-sweetened beverages (SSB), and milk. **Results:** 87% of Brazilians aged 10 and older consumed coffee on a giving day. The consumption of all food groups decreased as days of coffee consumption increased, except for non-caloric sweeteners and sugar, which increased. In the case-crossover analysis (2192 men and 2580 women), women who consumed coffee on one of the two days showed an increase of 10 g of sugar and of 0.10 mL (2 drops) of sweeteners. For men values were 8 g and 0.05 mL. Also, women reduced sugar-sweetened beverages (−56.8 mL/day), while men reduced milk intake (−25.9 mL/day). **Conclusions:** Coffee consumption was associated with increased intake of sugar and non-caloric sweeteners and lower intake of sweets, SSB, and milk. Reducing sweets and SSB is beneficial but increasing sweeteners and reducing milk are not. Potential strategies include encouraging the use of milk in coffee instead of sugar and sweeteners, as well as reducing the size of sugar sachets, which in Brazil typically range from 5 g to 8 g.

## 1. Introduction

We are analyzing population-based data using a case-crossover design to evaluate companion food items associated with coffee intake. This design, commonly used in other fields, offers the significant advantage of controlling for all confounding factors by comparing individuals with themselves.

Coffee is one of the most consumed beverages worldwide. In the Americas and Europe, it is widely preferred over tea, with Brazil ranking as the second-largest global consumer [1]. The caffeine in coffee offers stimulating properties that can enhance memory, attention, and concentration [2,3,4]. Moderate consumption of 3 to 4 cups daily is associated with a reduced risk of mortality from all causes and cardiovascular diseases [5]. In Brazil, a 2021 survey revealed that 34% of coffee consumers prefer the drink without sugar, 39% consume it with sugar, 19% with milk and sugar, and 10% choose coffee with milk without sugar [1].

Although coffee is highly appreciated, the addition of sugar or artificial sweeteners can compromise the health benefits the beverage offers. The consumption of sugary drinks is a growing public health concern due to associated risks, such as hormonal dysregulation, insulin resistance, dyslipidemia, obesity, and dental caries [6,7]. Moreover, the sugar in beverages is low in nutritional value, high in calories, and provides minimal satiety, contributing to weight gain and the onset of chronic conditions such as obesity, type 2 diabetes, and cardiovascular diseases [8,9,10,11,12]. Consequently, reducing added sugar in beverages is a key recommendation in nutritional guidelines [13,14,15].

The relationship between coffee consumption and sugar use is particularly relevant in the Brazilian context. The Longitudinal Study of Adult Health (ELSA-Brasil) establishes that sweetened coffee is the most consumed beverage in the country, with sugar being the predominant method of sweetening coffee [16]. Similarly, the 2017-18 National Food Survey reveals that coffee is one of the most consumed foods in the country [17]. Although coffee offers proven health benefits, the addition of sugar can diminish these positive effects. A cohort study examined the relationship between sugar-sweetened coffee, artificially sweetened coffee, and unsweetened coffee with all-cause mortality. After seven years of follow-up, the results indicated that only individuals who consumed unsweetened coffee experienced a reduced risk of all-cause mortality, highlighting the benefits of unsweetened coffee consumption [18].

In addition to sugar, artificial sweeteners also warrant attention. The World Health Organization (WHO) has highlighted the potential adverse effects of prolonged non-caloric sweetener (NCS) use, including an increased risk of type 2 diabetes, cardiovascular disease, and mortality in adults. Research suggests that replacing sugar with artificial sweeteners has not been effective for long-term weight management, raising concerns about the perceived benefits of these seemingly healthier food alternatives [19]. The use of artificial sweeteners, particularly in beverages like coffee, has been widely debated due to their potential harmful health effects, such as increased obesity and cardiometabolic diseases [20,21].

Coffee consumption may also influence the choice of other sweet foods, possibly due to caffeine’s influence on appetite and food preferences [3,4]. Arruda et al. (2009) suggest that replacing coffee with milk, chocolate, or soft drinks during breakfast or an afternoon is a common practice among Brazilian adults, which may negatively impact total food intake [22].

On the other hand, adding cow’s milk to coffee has nutritional benefits, as milk is a valuable source of protein, vitamins, and calcium [23]. However, only 30% of Brazilian adults consume milk daily [17]. A study by Min et al. (2023) found that among adults, the consumption of ground coffee or coffee with milk was associated with a lower risk of developing depression and anxiety [24].

This study’s originality lies in its use of a case-crossover design, which allows for understanding how coffee consumption influences the intake of other food groups. By comparing individuals with themselves, this design effectively controls all potential confounding factors. The results can inform campaigns aimed at reducing unhealthy dietary patterns associated with coffee consumption. Additionally, the study is based on national data from the National Food Survey (INA), a subsample of the 2017–2018 Household Budget Survey (POF). This enables a detailed and contextualized analysis of individual dietary intake within the Brazilian population [25].

Thus, the study aimed to investigate the relationship between coffee consumption, added sugar, non-caloric sweeteners, and other related food groups using data from a large, nationwide Brazilian survey.

## 2. Materials and Methods

### 2.1. Study Population

This study was conducted using data from the population participating in the 2017–2018 edition of the National Food Survey (INA), a module of the Household Budget Survey (POF), a nationwide survey. The POF collects information on the family expenses, living conditions, and consumption habits of Brazilian families. Detailed information about the sampling procedure is available in the official publications of the Brazilian Institute of Geography and Statistics [25].

The INA collected two 24-h recalls of all residents aged 10 and over among 20,112 randomly selected households, a subsample equivalent to 34.7% of the 57,920 households investigated by the POF, providing information on 46,164 residents.

Data from the National Food Survey (INA) are open access and available on the IBGE website.

For the case-crossover analysis, participants who answered only one day of the 24-h recall were excluded. The final sample consisted of 38,854 participants.

### 2.2. Study Design

For the primary analysis, we employed a modified case-crossover design, comparing individuals’ dietary behaviors on days when they consumed coffee versus days when they did not. This approach was used to evaluate the impact of coffee consumption on sugar intake and other dietary components.

Figure 1 is the schematic presentation of the cross-over design. Subjects 2 and 3 contributed to the analysis.

### 2.3. Individual Food Consumption

The 24-h dietary recalls were conducted on two non-consecutive days selected throughout the week, during which the research agent was present in the household. Data collection was performed using the Automated Multiple-Pass Method [26]. In the first phase, interviewers instructed participants to record everything they had consumed the day before the interview, quickly noting it on paper. In the second phase, information about portion sizes, preparation methods, and whether the foods were consumed at home or outside were recorded in specific software which includes 1832 items, but new ones could be added. The nutrient intake was based on version 7.0 of the Brazilian Table of Food Composition (TBCA) from the Food Research Center at the University of São Paulo [27].

The Table of household food measurements of Brazil was used to estimate the quantity consumed in grams or milliliters of each food or drink. This table was developed in the previous edition of the POF (2008–2009) and was later revised and updated in the POF 2017–2018 [28].

Participants provided information on the use of items that are normally added to selected foods and drinks, including table sugars. The amount of sugar added to coffee or other food items was estimated as 10% of reported food consumption [25].

### 2.4. Food Groups

The food items included in this study were those that could be consumed with coffee or exchanged for coffee in Brazil (Table 1).

### 2.5. Statistical Analysis

The main analysis compared food items on coffee consumption and non-consumption days for the same individuals, with differences in the quantity of food consumed calculated. The food groups included milk, sandwiches, sweets, sugar-sweetened beverages (SSBs), chocolate, sugar, and non-caloric sweeteners. The differences between coffee consumption and non-consumption days were calculated individually.

Additionally, to provide an overview of food group consumption in the population aged 10 years or older, the averages and consumption frequencies were estimated according to coffee intake days and by sex.

The results were presented with 95% confidence intervals. All analyses were conducted using the SAS OnDemand for Academics statistical package (SAS Institute Inc., Cary, NC, USA), considering the weighting and complexity of the sampling plan from the 2017–2018 Household Budget Survey (POF). This ensured the representativeness of the results for the Brazilian population. The dataset underwent a series of descriptive and inferential analyses, which assessed the differences in food and beverage consumption.

## 3. Results

In Brazil, 87% of the population aged 10 years or more reported coffee consumption in given day. Regardless of coffee consumption or not, the daily frequency of reporting the use of added sugar was 80%, for non-caloric sweeteners was 8.6%, and the use of both caloric and non-caloric sweeteners was 5.3%. As the frequency of coffee consumption increased the frequency of sugar intake increased from 75.6% (95% CI 73.1–78.1) for no coffee consumption in both days to 80.4% (95% CI 79.3–81.4) for coffee drinking in both days (Table 2).

Table 3 describes the average consumption of food groups by sex, according to the frequency of coffee consumption (no day, one day or two days). There was a downward trend in the consumption of all food groups as the number of days of coffee consumption increased in both sexes. Men consumed more sugary drinks than women in all categories, while women consumed more milk. Consumption of sweets was slightly higher among women when they do not drink coffee, but both sexes show a similar reduction in consumption as the number of days they drink coffee increases. Consumption of chocolate and chocolate products was similar between sexes (Table 3).

The main analysis was based on the case crossover design (Table 4). Mean difference in food consumption between days with and without coffee consumption.

There was an association between coffee consumption and consumption of food groups. For both sexes, there was a statistically significant increase in the consumption of sugar and non-caloric sweeteners and a significant reduction in the consumption of sugary drinks and milk on days when people drank coffee. Women tended to consume less chocolate and chocolate products on days when they drank coffee (−1.87; IC 95%: −3.40 to −0.34). This difference compared to men was statistically significant since confidence interval does not overlap (0.31; IC 95%: −0.34 to −0.96). For non-caloric sweeteners greater difference was observed for women (Table 4).

## 4. Discussion

The results of this study showed an association between coffee consumption and both positive and negative dietary habits in Brazil. The analysis revealed that 87% of the population over 10 years of age consumed coffee, with a higher prevalence of sugar consumption among coffee consumers, but the estimated amount of sugar related to coffee consumption was small, approximately 10 g, indicating a modest impact on sugar intake. However, the association between coffee and increased consumption of non-caloric sweeteners raises concerns, since these ingredients may attenuate the potential health benefits of coffee, with women showing great difference compared to men.

Previous studies, such as ELSA-Brasil, highlight that sweetened coffee is the most consumed beverage by Brazilians, regardless of age group, and added sugar is the main method for sweetening beverages [29]. Excessive consumption of sugar and artificial sweeteners contributes to increased risks of obesity, type 2 diabetes, and other chronic diseases [8,9].

Per capita coffee intake in the survey was 163 g per day [25], and each cup of coffee (87 g) was usually served with two sachets of sugar (10 g). Thus, for 80% of the population, those who add sugar to their coffee, the sugar intake associated with coffee consumption was close to 20 g per day. Comparing days with coffee intake with days without coffee intake, the difference in sugar intake was 10 g for men and 8.4 g for women, so coffee intake is not the main source of sugar intake. Although the increase in sugar intake with coffee is small, it still deserves attention due to its possible long-term health impacts, especially considering the high prevalence of coffee consumption with sugar in Brazil. In this context, public health campaigns could encourage the consumption of coffee without sugar or sweeteners. Public health campaign could encourage the use of milk in coffee instead of sugar and sweeteners, as well as promoting reduction in the size of sugar sachets, which in Brazil typically range from 5 g to 8 g.

In this survey almost 10% of the population reported using non-caloric sweeteners, varying from 6.9 among adults, 20- to 60-year-old, to 19.9% among the elderly [30]. Also, on those days with coffee intake, compared to days without, difference doubled among women indicating coffee as an important source of non-caloric sweeteners. The association of sugar-sweetened and artificially sweetened beverages with obesity, cardiometabolic diseases and other chronic diseases raises concerns about the addition of sugar or artificial sweeteners to coffee [20,21]. A prospective cohort study of 146,566 Korean adults reported that consumption of ground coffee, coffee with milk and coffee without sugar was associated with a lower risk of incident depression and anxiety [22].

Another important finding was the negative association between coffee and milk consumption, in a country where milk consumption is low, only 30% of adults drink milk daily [17]. The daily intake of milk among men was 56.7 mL on days without coffee consumption and 13 mL when coffee was reported on both days, among women these values changed from 62.4 mL to 17 mL. Thus, in Brazil a regular coffee consumer had a negative impact on milk consumption, which represents a nutritional concern, since milk is an important source of calcium and other essential nutrients [23]. The competition between coffee and milk consumption in the Brazilian diet points to a need for health policies that promote milk consumption or the inclusion of alternatives that can compensate for this reduction.

Although we observed that sugar intake increased as coffee consumption frequency increased, this change is very small. Furthermore, coffee drinkers of both sexes were less likely to drink sugar-sweetened beverages. An estimated reduction of approximately 50 mL of SSB was associated with coffee intake. The reduction in sugar-sweetened beverage intake is a significant finding, as it suggests that coffee may function as a substitute beverage with potential public health use.

Globally, there is a clear link between the consumption of sugar-sweetened beverages and several adverse health conditions, such as obesity, type 2 diabetes, and cardiovascular disease [8,9,10,11].

Comparing data from the POF (2008–2009) and POF (2017–2018), the use of table sugar decreased by 8%, while the habit of not using any sweetener increased almost threefold [10]. According to the World Health Organization (WHO), the use of sugar-free sweeteners (NSS) can cause potentially undesirable effects of long-term use, such as increased risk of type 2 diabetes, cardiovascular disease, and mortality in adults. Furthermore, replacing free sugars with NSS does not improve long-term weight control [27,31].

Stratified analyses by sex indicated that women reduced more their consumption of chocolate, compared to men. Women typically report higher consumption of sugary items, including chocolate [17]. The reduction in chocolate intake among women was unexpected and should be explored in further studies.

The differences presented in this study for food consumption on days when individuals drink and do not drink coffee suggest possible strategies to reduce unhealthy additions to coffee consumption. This is an important public health policy, given the many positive aspects of coffee itself.

The strengths of this study include its case-crossover design of a large population-based study, allowing full control of confounding factors and providing a snapshot of dietary relationships in the population. As a limitation, data were based on participant reports, which may introduce bias. The main bias in the 24-h recall is underestimation. In Brazilian surveys, this underreporting may correspond to approximately 20% of energy intake [32]. However, the 24-h recall is the better method to evaluate population food intake, although it does not reflect the individual’s habitual dietary pattern, but rather the intake of the previous or current day. However, it is the best choice to estimate the average consumption in population groups when applied to a representative sample of the population on different days of the week, as was done in this investigation.

## 5. Conclusions

It has been observed that, although coffee is widely consumed and has recognized benefits, it is often associated with an increase in the consumption of sugar and non-caloric sweeteners, which may mitigate its positive effects.

On the other hand, coffee demonstrated a positive substitution effect by being associated with a reduction in the consumption of sugary drinks and sweets, especially among women.

However, the decrease in milk consumption on days with coffee intake is a concern, since milk is an important source of essential nutrients, such as calcium and protein. This decline may have long-term nutritional implications, especially in a population where daily milk consumption is already low.

Strategies that encourage the addition of milk to coffee instead of sugar or artificial sweeteners, as well as exploring the potential of coffee as a substitute for other sugary beverages, represent important avenues for public health policies that seek to improve the population’s dietary quality and reduce the risks associated with excessive consumption of sugar and sweeteners. Therefore, future public health campaigns could focus on promoting coffee without sweeteners or with healthier alternatives, such as small amounts of milk. This approach is relevant not only at the national level but also globally, as coffee consumption is widespread across many cultures, and the health issues linked to excessive sugar and artificial sweetener intake are a global concern.

## Figures and Tables

**Figure 1 nutrients-16-04019-f001:**
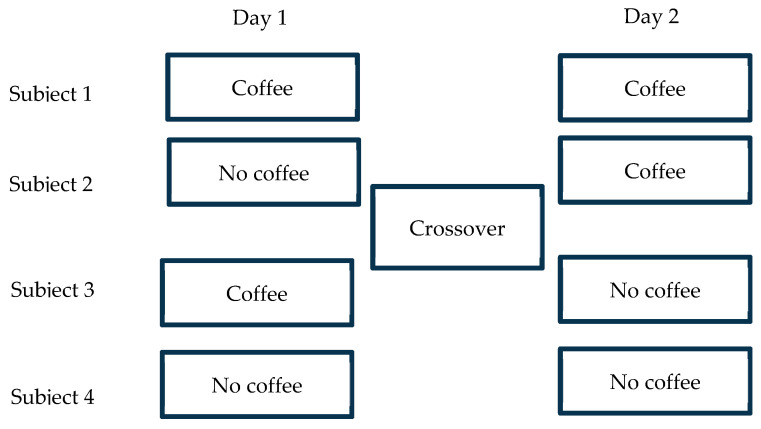
Schematic presentation of crossover design.

**Table 1 nutrients-16-04019-t001:** Food groups and items analyzed.

Food Groups	Food Items
Coffee	coffee, espresso coffee, carioca coffee, decaffeinated coffee, coffee with milk, cappuccino coffee, decaffeinated cappuccino coffee, light/diet cappuccino coffee, coffee with milk substitute and decaffeinated espresso coffee
Sugar	sugar, demerara sugar and brown sugar
Non-caloric sweeteners	light sweetener, artificial sweetener, light sugar, light powdered sweetener and light liquid sweetener
Sugar Sweetened beverages	Soft drinks, industrialized soft drinks/juices, juices
Sweets	Sweet breads, cakes, filled cakes, sweet cookies, filled cookies, milk-based sweets, peanut-based sweets, fruit-based sweets, ice cream/popsicles
Milk	Whole milk, skimmed milk, milk-based preparations
Chocolates and chocolate products	Chocolate bars, tablets, confetti, powder, cream, syrup, bonbons, truffles and fillings. Chocolate powder and ovomaltine.

**Table 2 nutrients-16-04019-t002:** Frequency of added sugar and non-caloric sweetener use (%), with (95% confidence interval), according to frequency of coffee intake (n = 38,854).

		Coffee				
	No Consumption (n = 5139)		1 Day (n = 4772)		2 Days (n = 28,943)	
	%	CI (95%)	%	CI (95%)	%	CI (95%)
Sugar	75.6	73.1–78.1	78.7	76.6–80.9	80.4	79.3–81.4
Non-caloric sweeteners	8.6	6.3–10.5	7.7	6.5–8.9	8.8	8.2–9.5
Sugar and sweetener	4.9	3.8–6.1	5.5	4.4–6.7	5.4	4.8–5.9
Not used	11.1	9.3–12.8	8.1	6.4–9.7	5.5	4.7–6.2

**Table 3 nutrients-16-04019-t003:** Average intake (g or mL) of food groups (sweetened beverages, sweets, milk and chocolates), according to days of coffee consumption, and sex.

**Men**			
**Food Groups**	**Mean**		
	**No Consumption** **(n = 2385)**	**1 Day** **(n = 2192)**	**2 Days** **(n = 13,392)**
Sweetened drinks	325.7	290.1	202.0
Sweets	42.3	37.4	32.6
Milk	56.7	31.2	13.1
Chocolate	4.62	1.40	1.05
**Women**			
**Food Groups**	**Mean**		
	**No Consumption** **(n = 2754)**	**1 Day** **(n = 2580)**	**2 Days** **(n = 15,551)**
Sweetened drinks	271.6	210.9	163.0
Sweets	45.2	38.6	32.0
Milk	62.4	33.0	17.2
Chocolate	4.90	2.26	1.24

**Table 4 nutrients-16-04019-t004:** Difference (g or mL) and confidence interval (CI) between the amount of food consumed in the day each individual drinks coffee, compared to non-drinking coffee, by sex.

Food Groups	Men (n = 2192)			Women (n = 2580)		
	Mean	CI (95%)		Mean	CI (95%)	
Sugar	10.1	8.1	12.0	8.38	6.61	10.2
Non-caloric sweeteners	0.05	0.03	0.07	0.10	0.07	0.13
Sweetened drinks	−47.7	−74.1	−21.2	−56.8	−73.7	−39.9
Sweets	−2.7	−8.79	3.39	−0.65	−4.71	−6.00
Milk	−25.9	−33.7	−18.2	−21.6	−28.5	−14.7
Chocolate	−0.31	−0.34	−0.96	−1.87	−3.40	−0.34

## Data Availability

Data files and documentation from the surveys are publicly available from the Brazilian Institute of Geography and Statistics at https://www.ibge.gov.br/estatisticas/sociais/trabalho/9050-pesquisa-de-orcamentos-familiares.html?edicao=9061&t=downloads (accessed on 12 May 2024).

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
