# Peer review of "Coffee Intake in Brazil Influences the Consumption of Sugar, Sweets, and Beverages"

_nutrients, 2024, doi:10.3390/nu16234019_

Round 1
Reviewer 1 Report
Comments and Suggestions for Authors
Dear authors,
The topic of the article is extremely interesting. The abstract should be rewritten so that the authors avoid explicitly mentioning its main parts, such as: purpose, methodology, results, etc. While it is very good that the abstract is structured in this way, there is no need to label these subsections. Additionally, a section dedicated to the managerial and theoretical implications of the research is missing at the end of the abstract. The introduction is extremely superficial. The authors should separate it into two subsections: the first for the introduction and the second for the literature review. The introduction must clearly articulate the research problem and the gap identified in the literature by the authors. Additionally, the purpose of the research should be clearly stated. Regarding the literature review, it needs to include many more studies conducted at the international level that relate to the researched topic. The authors are requested to provide information regarding the open access to the secondary data used in the study. The statistical analyses performed are not detailed enough for the study to be replicated. The results are extremely limited, and the discussions need to be improved by correlating them with the literature review. The conclusions are very superficial, consisting of only four lines.
Reviewer 2 Report
Comments and Suggestions for Authors
1.In Figure 1, please describe what does the “A”, “B”, “C”, “D” mean? And what would they be treated in the analysis.
2.Please elucidate that how the “non-consecutive days” might have something to cofound the research? Such as which two days were decided? Was it be decided randomly? How long between the two days, its mean and deviation. Was the survey be influenced by different seasons?... Maybe they can be discussed or reported in limitation of this research.
3.For the title of Table 4, “Difference (g or mL) and confidence interval (CI) between the amount of food consumed in the day each individual drinks, compared to non-drinking coffee, by gender.” It would be better to add a word “coffee” to become “Difference (g or mL) and confidence interval (CI) between the amount of food consumed in the day each individual drinks coffee, compared to non-drinking coffee, by gender,” that would be more understandable.
4.In abstract, “…who drank coffee on one of the two days, showed an increase in sugar consumption of about 10 g, an increase in non-caloric sweeteners of 0.10 ml (2 drops), comparing days with and days without intake of coffee” seemed not consistent with Table 4. Who compared with who?
5.section 2.2 “study design“ and Figure 1 need more explanation to help understand the Result section.
6. Some “food category” terms should be used consistent in the manuscript to avoid misunderstandings, such that the food groups in Table 2 and Table 3 were not consistent.
Reviewer 3 Report
Comments and Suggestions for Authors
You should start your abstract with a background statement to justify the need for carrying out your study and then present the study’s main goals.
Line 29: Please, revise this: “n in milk are not. trategies”.
Mention future perspective at the of your abstract.
In the introductory section, you have to show a worldwide perspective of the topics that will be addressed in your study and cite more studies addressing this.
Why did you only apply the 24-hour dietary recalls in such a wide study population? More efficient tools should be considered.
The results section is fine but more discussion should be provided from a worldwide perspective and the study limitations have to be addressed in a more depth way. In my point of view, the use of only 24-hour dietary recalls is a very relevant drawback.
Conclusions have also to be expanded and better elaborated. What can we learn from this study (international scientific community and population)? Directions for further research are missing.
Sincerely, I believe that the manuscript in its current form is not adequate to be published in an international Q1 journal like Nutrients and the authors should consider applying other techniques and not only the 24-hour dietary recalls.
Round 2
Reviewer 1 Report
Comments and Suggestions for Authors
The authors have responded to the reviewer’s observations/comments only to a very limited extent. Under these conditions, the decision remains the same until the necessary revisions are made.
Reviewer 2 Report
Comments and Suggestions for Authors
OK
Author Response
Thank you very much for taking the time to review this manuscript.
Reviewer 3 Report
Comments and Suggestions for Authors
I continue to find it quite limiting and suspicious to publish in an international Q1 journal like Nutrients a study based solely on the use of 24-hour dietary recalls.
